# Cardiac Involvement in Patients with Multisystem Inflammatory Syndrome in Children (MIS-C) in Poland

**DOI:** 10.3390/biomedicines11051251

**Published:** 2023-04-23

**Authors:** Kamila M. Ludwikowska, Nafeesa Moksud, Paweł Tracewski, Mateusz Sokolski, Leszek Szenborn

**Affiliations:** 1Department of Pediatric Infectious Diseases, Wroclaw Medical University, Ludwika Pasteura 1, 50-367 Wrocław, Poland; 2Laboratory of Genetics and Epigenetics of Human Diseases, Hirszfeld Institute of Immunology and Experimental Therapy, Polish Academy of Sciences, Rudolfa Weigla 12, 53-114 Wrocław, Poland; 3Department of Pediatric Cardiology, Regional Specialist Hospital in Wroclaw, Research and Development Center, Kamieńskiego 73a, 51-124 Wrocław, Poland; 4Institute of Heart Diseases, Wroclaw Medical University, Borowska 213, 50-556 Wroclaw, Poland

**Keywords:** paediatric inflammatory multisystem syndrome temporally associated with SARS-CoV-2, PIMS-TS, acute heart failure, children, takotsubo syndrome, COVID-19, ejection fraction, LVEF, coronary artery aneurysm, CAA, Kawasaki disease

## Abstract

Multisystem inflammatory syndrome in children (MIS-C) is an immune-mediated complication of severe acute respiratory syndrome coronavirus 2 (SARS-CoV-2) infection. Cardiovascular system is commonly involved. Acute heart failure (AHF) is the most severe complication of MIS-C, leading to cardiogenic shock. The aim of the study was to characterise the course of MIS-C with a focus on cardiovascular involvement, based on echocardiographic (echo) evaluation, in 498 children (median age 8.3 years, 63% boys) hospitalised in 50 cities in Poland. Among them, 456 (91.5%) had cardiovascular system involvement: 190 (48.2%) of patients had (most commonly atrioventricular) valvular insufficiency, 155 (41.0%) had contractility abnormalities and 132 (35.6%) had decreased left ventricular ejection fraction (LVEF < 55%). Most of these abnormalities improved within a few days. Analysis of the results obtained from two echo descriptions (a median of 5 days apart) revealed a >10% increase in LVEF even in children with primarily normal LVEF. Lower levels of lymphocytes, platelets and sodium and higher levels of inflammatory markers on admission were significantly more common among older children with contractility dysfunction, while younger children developed coronary artery abnormality (CAA) more often. The incidence of ventricular dysfunction might be underestimated. The majority of children with AHF improved significantly within a few days. CAAs were relatively rare. Children with impaired contractility as well as other cardiac abnormalities differed significantly from children without such conditions. Due to the exploratory nature of this study, these findings should be confirmed in further studies.

## 1. Introduction

Cardiac complications affect a significant percentage (from ~7 to >30%) of patients of all ages who are infected with severe acute respiratory syndrome coronavirus 2 (SARS-CoV-2) infection [1]. It became clear that a history of cardiovascular disease is a risk factor for a severe course of coronavirus disease 2019 (COVID-19) soon after the pandemic began [2]. Further observations established that SARS-CoV-2 infection may be a risk factor for cardiac complications as well, even in previously healthy patients [3,4,5,6,7,8,9,10,11]. The most intriguing cardiological phenomena associated with the COVID-19 pandemic include cases of acute heart damage in patients who have multisystem inflammatory syndrome in children (MIS-C) [12,13,14]. This new disease appeared during the COVID-19 pandemic in children, most commonly of school age [14,15,16,17]. Isolated cases in young or middle-aged adults are referred to as multisystem inflammatory syndrome in adults (MIS-A) [18,19,20].

MIS-C is a late, immune-mediated complication occurring about 4 weeks after SARS-CoV-2 infection in 1 in 3000 unvaccinated children [21,22,23]. Due to the similarity of their clinical symptoms, MIS-C is compared with Kawasaki disease (KD), toxic shock, syndrome (TSS) and macrophage activation syndrome (MAS). Recently, however, it has become known as a separate and new disease entity [24,25,26]. The exact pathogenesis of changes in the course of MIS-C, including cardiac involvement, is yet to be known [22].

MIS-C includes gastrointestinal, mucocutaneous and neurological symptoms, as well as cardiac disorders. The most frequently observed cardiac abnormalities include ventricular dysfunctions, which may lead to cardiogenic shock. Pericardial effusion or coronary artery dilatation and aneurysms (CAAs) occur less frequently [12]. The acute and reversible nature of the heart failure in the course of MIS-C might resemble the picture of takotsubo syndrome (TTS) [27,28]. We have described such a case of TTS in the course of MIS-C before [29].

More precise characteristics of cardiovascular involvement come mostly from single-canter observation studies of relatively small groups of patients [14,30,31,32,33,34]. Regardless, the one undoubted conclusion from all the previous reports is that the involvement of the cardiovascular system is one of the most important features of MIS-C, and a paediatric cardiologist should be involved in the diagnostic and therapeutic process [35] to ensure optimal patient care.

The MultiOrgan Inflammatory Syndromes COVID-19 Related Study (MOIS-CoR) is a register of inflammatory diseases in children that captures and characterizes MIS-C cases in Poland [36,37]. Every wave of COVID-19 was followed by a wave of MIS-C cases of variable intensity. More than five hundred cases of MIS-C, defined according to the World Health Organization (WHO) [38], were reported by April 2022 [36]. Many paediatric centres struggled with limited access to paediatric cardiologists, as there are less than 200 such specialists in the country of 37 million people, including about 7 million children [39].

The main aims of our study were to (i) characterize cardiovascular involvement in MIS-C in our population and (ii) identify patients with MIS-C who should be prioritised for highly specialised cardiac referral and diagnostics. We specifically aimed to identify the differences between children with and without contraction dysfunction or CAAs and to investigate the incidence of TTS echocardiographic signs in the course of MIS-C in our population.

## 2. Materials and Methods

### 2.1. Study Sample

A detailed description of the MOIS-CoR Study was presented in our previous publication [37]. In this retrospective case series study, we report data covering the period from 4 March 2020 (when the first COVID-19 case was confirmed in Poland) to 14 April 2022, coming from 50 Polish cities. We concentrated particularly on cardiovascular involvement in MIS-C. Therefore, we only included children who:(1)Fulfilled the criteria for MIS-C adapted from the WHO;(2)Did not have any previous cardiovascular comorbidities (including congenital heart defects or hypertension);(3)Had at least one echocardiography (echo) result available (Figure 1).

### 2.2. Study Design and Definitions

Anonymised patient data including demographic, clinical characteristics, laboratory test results, cardiovascular evaluation results and treatment data were analysed. Vital signs and laboratory parameters were obtained at admission and at their respective peaks. Body mass index (BMI) was converted into Z-scores based on the WHO reference standards for children younger than 5 years [40] and national reference standards for older children [41].

For the echocardiographic findings, we extracted information on:(1)Contractility abnormalities—the presence of any contractility dysfunction described by the cardiologist.(2)Valvular insufficiency—graded based on the interpreting physician’s qualitative assessment.(3)Left ventricular ejection fraction (LVEF)—evaluated using Simpson’s biplane method as normal (LVEF ≥ 55%) or decreased (LVEF < 55%).(4)Pericardial effusion—as graded by the interpreting physician’s assessment.(5)Coronary arteries abnormalities (CAAs)—CAAs were further categorised based on the Z-scores according to Dallaire [42] into dilation (a Z-score between 2 and 2.5), small aneurysms (a Z-score between 2.5 and 5), medium aneurysms (a Z-score between 5 and 10 and absolute dimension less than 8 mm) and large aneurysms (a Z-score ≥ 10 or absolute dimension ≥ 8 mm) [43]. If there were no precise measurements of the coronary artery diameter, but a CAA was diagnosed, it was categorised either as dilation or an aneurysm according to the description.

If a full description of echocardiographic findings was not available but information on normal results was provided, we categorised it as absence of listed abnormalities accordingly.

We introduced study definitions as follows:A.Laboratory findings for heart involvement—the presence of any of the listed abnormalities during the course of the hospitalisation:
○Elevated troponin (in respect to the type of troponin measured and the laboratory norm of the reporting site);○Concentration of brain natriuretic peptide (BNP) ≥ 100 pg/mL;○Concentration of N-terminal prohormone for brain natriuretic peptide (NT-proBNP) ≥ 300 pg/mL.B.Echocardiological findings for heart involvement—the presence of any of the listed abnormalities in any of the available echo results during hospitalisation:
○CAAs;○Decreased LVEF;○Contractility abnormalities;○Valvular insufficiency (aortic valve insufficiency, IAoV; mitral valve insufficiency, IMV; pulmonary valve insufficiency, IPV; or tricuspid valve insufficiency, ITV);○Pericardial effusion.

To determine the predictors and markers of echocardiographic abnormalities, we divided the study population into clusters, as presented on Figure 2.

### 2.3. Statistical Methods

Variables were described in relation to the sum of non-missing observations for which the variable was recorded. Categorical variables were represented as counts and as percentages of non-missing values. Continuous data were represented as medians with upper (75th) and lower (25th) quartiles. Data were evaluated for normal distribution using density histograms overlaid with kernel density plots against a projected normal curve. Fisher’s exact test was used to assess independence between categorical variables. Student’s t-test and Kruskal–Wallis H test were used for categorical–continuous variable pairs. A significance level of 0.05 and two-sided testing were employed. We assumed missing values to be distributed randomly and independently from the data, and these missing data were treated through clinical definitions according to Łukasiewicz’s logic. A one-way analysis of variance (ANOVA) was used to compare our three major age groups (<5 years, 5–11 years and 12–18 years) against potential covariates that may vary with age. Bartlett’s test was calculated alongside the one-way ANOVA to check if variances were equal across groups or samples. To discern covariates that may predict echocardiological involvement in patients with MIS-C, patient characteristics, presenting symptoms and laboratory findings on admission were compared in two patient populations: patients with negative echocardiographic findings (consisting of patients who have normal cardiac lab markers with normal echo findings and patients with elevated cardiac lab markers) versus patients with positive echocardiographic findings (irrespective of having elevated or normal cardiac lab markers). Initial and follow-up echocardiographic characteristics of the overall cohort were evaluated using the abovementioned statistical methods. The subset of patients who had both initial and follow-up echo results were compared using McNemar’s test for categorical variables, and by paired-t-tests for continuous variables. Additionally, comparisons between patients with normal and reduced LVEF were made for both initial and follow-up echo findings. *p*-values were also calculated between initial and follow-up echo findings in patients with normal and reduced LVEF. To further assess cardiac involvement by specific cardiac abnormalities, variables such as patient demographics, laboratory and vital signs at admission and presenting symptoms were tested against study populations, defined as patients with and without CAAs at any time; patients with and without pericardial effusion at any time; patients with and without reduced LVEF at any time; and patients with and without contractility abnormalities at any time. All statistical analyses were performed in Stata 16.1 (College Station, TX, USA: StataCorp LLC.). MIS-C incidence was estimated based on the demographic data published by Statistics Poland.

### 2.4. Ethical Statement

Ethical approval was obtained from the Bioethics Committee at Wroclaw Medical University (CWN UMW BW: 313/2020). Waiver of informed consent was obtained with only deidentified data transmitted and analysed.

## 3. Results

### 3.1. Study Sample

Four hundred ninety-eight children (median age 8.3 years, 63% male, all of White ethnicity except for two Asian patients) were included in the analysis (Figure 1). Ninety percent of children were previously healthy, with the most common comorbidity being asthma (5%). Seven percent were obese. The characteristics of the study group and MIS-C presentation depending on age group (<5; 5–11; 12–18 years of age) are presented in Table 1 and Appendix A. Gastrointestinal (GI), mucocutaneous, neurological and cardiovascular systems were the most commonly involved overall. Clinical presentation differed by age with GI, with lower respiratory symptoms and osteoarticular and muscle involvement appearing more commonly in older children. A significant proportion of patients had abnormal vital signs. Overall, 49% of children developed hypotension, which was significantly more common in older age groups. The laboratory parameters of MIS-C encompassed lymphopenia, high concentrations of inflammatory markers such as C-reactive protein (CRP) and procalcitonin, hyponatremia, hypoalbuminemia, elevated concentration of D-dimers and BNP/NT-proBNP and less often, elevated troponin. There were significant differences between age groups in laboratory test results as well. Older children had lower lymphocyte counts, higher CRP and ferritin concentrations, lower sodium concentrations, more frequent occurrences of renal function impairment and elevated troponin.

A total of 93% of patients had laboratory findings for heart involvement and 56.2% had echocardiographic findings for heart involvement.

### 3.2. Laboratory Markers for Cardiovascular Involvement

The majority of children had elevated levels of either BNP/NT-proBNP or troponin (Figure 2). Median BNP concentration on admission was 287 pg/mL (IQR 114–1006) and that of NT-proBNP was 2176 pg/mL (IQR 486–7526). Median values at the respective peak are presented in Table 1 and Appendix A. A total of 63 (26%) children had elevated troponin on admission and 100 children had elevated troponin (38%) at any point during the hospitalisation.

### 3.3. Echocardiography Findings

All children included in this analysis had an echo performed at least once during their hospitalisation. Median time from the disease symptoms onset to hospital admission was five (IQR 4–6) days and that to the initial echo examination was six (IQR 4–8) days. For 337 (68%) children, a follow-up echo was performed with a median of five (IQR 2–7) days passing from the initial echo and a median of 11 (IQR 8–14) days from the disease onset. Median hospital stay was 11 (IQR: 9–14) days.

The most common echo finding was valvular insufficiency, which was present in 48% of children. Atrioventricular valve insufficiency was the most common type, with mitral valve insufficiency (IMV) described in 43% and tricuspid valve insufficiency (ITV) in 19% of children. Pulmonary valve insufficiency (IPV) was described in 3% and aortic valve insufficiency (IAoV) only in three patients. Most of the cases of regurgitation were considered benign; 10% of children had valvular insufficiency assessed as moderate or severe at the initial echo and 8% in the follow-up echo (Table 2 and Appendix A).

The second most common echo finding was contractility abnormality, which was described in 41% of children at any time of the hospitalisation, with decreased LVEF in 36%. Median LVEF was 55% (IQR 17) in the initial echo and 56% (IQR 17) in the follow-up echo. Twenty-six percent of patients had decreased LVEF in the initial echo and 14% in the follow-up echo (Table 2 and Appendix A).

CAAs (aneurysm or dilation) were found in 11% of children (8 dilations, 11 small, 5 medium, 1 large aneurysm and 1 of unspecified size in the initial echo, and 6 dilations, 8 small, 3 medium and 2 large aneurysms in the follow-up echo). CAAs were found significantly more often in younger age groups. The opposite was true for contractility abnormalities and decreased LVEF, which were more common in older age groups, as seen in Table 1 and Appendix A.

By the time of the follow-up echo, a significant improvement was noted for valve insufficiency, contractility abnormalities and LVEF. We did not find significant differences in CAAs and pericardial effusion between the initial and follow-up echoes (Table 2 and Appendix A). Interestingly, there were more children with segmental contraction abnormalities including apex rounding in the follow-up echo (20 cases) than in the initial one (11 cases), but the general number of children with this condition during hospitalisation was relatively low (31 cases).

The baseline median LVEF increased significantly from 56% (IQR 47–64) to 57% (IQR 50–69) in the follow-up echo (*p* < 0.01). Cases with an increase over 10% in LVEF was observed even in children who had a baseline echo described as normal (with LVEF > 55%; Figure 3).

Ejection fraction (EF)—left ventricular ejection fraction was evaluated using Simpson’s biplane method. The blue line indicates the value of 55%, which is considered as the norm for children.

Demographic and clinical features (signs and symptoms, laboratory test results and vital signs reported on admission) of children with and without echo findings were compared. From the 463 patients with laboratory and echo variables available, 264 children had any of the listed echo findings diagnosed. One hundred twenty-one patients had first and follow-up echoes performed and no echo abnormalities (Figure 2). Children with echo abnormalities more often presented neurological symptoms (including headache, lethargy or agitation) and breathing effort. They had lower systolic blood pressure and more often hypotension, and a higher respiratory rate on admission than children without any echo findings. Except for elevated troponin, they also had a higher procalcitonin concentration and lower sodium concentration (Table 3 and Appendix A).

A similar comparison was performed between children with and without particular echo findings (Table 4 and Appendix A).

We found that younger age, arthritis, conjunctivitis, eyelid swelling, a higher lymphocyte count and higher platelets on admission were predictive for CAAs. Children with abdominal pain, nausea or vomiting, conjunctivitis or swollen eyelids had CAAs less frequently than children without these signs.

Older age, the presence of neurological symptoms, breathing effort, chest pain, lower lymphocyte concentrations, lower platelet concentrations and higher CRP, higher procalcitonin, lower sodium, lower eGFR and higher BNP/NT-proBNP levels, as well as elevated troponin concentration on admission, were significantly more common in children with contractility dysfunction and decreased LVEF.

Children with contractility disorders and decreased LVEF more commonly had hypotension during their hospitalisation. Children with ventricular dysfunction required treatment at paediatric intensive care units more often.

Male sex and lower oxygen saturation were more often seen in children with pericardial effusion. Children with conjunctivitis, oral inflammation and diarrhoea had pericardial effusion less often.

## 4. Discussion

Several parallel definitions of MIS-C are used in the scientific literature [38,44,45]. In each of them, the diagnosis is based on clinical premises: fever, multiorgan damage including heart involvement, increased inflammatory markers and exclusion of other causes of the disease. Furthermore, they depend on epidemiological grounds, proven SARS-CoV-2 infection or contact with the virus preceding the disease. In our work, we adopted the WHO definition [38], which is the most rigorous one. Our study population did not differ significantly from those with MIS-C in previously published reports in terms of most common signs and symptoms as well as laboratory tests results [15,17,21,23,46].

The results of our study are in line with most of the previous findings and show that valvular insufficiency and myocardial dysfunction are the most common cardiological findings in children with MIS-C. The median LVEF in our study (55%) was comparable with the results presented in the meta-analysis of the studies concentrating on longitudinal cardiac outcomes in MIS-C (54%) [47]. IMV was slightly less common in our study (42.8% vs. 56.6%) and CAAs were definitively less common (10.5% vs. 23.7%) [47]. Such a discrepancy in the frequency of CAAs was noted before in our population [37] and may be explained by ethnic differences.

According to the previous reports, the most common cardiac abnormalities, which can be found in more than half of the patients with MIS-C, include features of acute myocardial injury in the form of mildly elevated cardiac troponins, significantly elevated BNP or NT-proBNP levels and echocardiographic features of myocardial contractility disorders, including reduced LVEF. Other possible cardiac disorders include acute valvular regurgitation, pericardial effusion, CAAs and disturbances in the electrophysiological function of the heart, such as arrhythmias, atrioventricular conduction disorders or changes in the ST segment and T wave [12,15,17,23,32,33,34,48,49]. The precise incidence of cardiac findings varies between previous reports and is hard to compare, as the authors used different, and sometimes not clearly specified definitions for cardiac parameters [15,17,23,46,49,50] or broad and unspecified study inclusion criteria [48]. More detailed cardiac characteristics come mostly from single-centre studies with relatively small study groups. To our knowledge, we present the largest cohort of patients with MIS-C characterised in terms of cardiac involvement so far.

Most of the studies presenting the acute phase of MIS-C concentrated on the values of LVEF at a single time point [15,17,32,46,49], while the studies with follow-ups presented the results from an interval of at least 2 weeks to a few months [30,47,51,52,53,54]. They all reported a significant improvement in children who had myocardial dysfunction. Our study supports these results. Within a median of five days, we found a significant reduction in the number of children with contraction abnormalities or decreased LVEF. Such a rapid improvement—within days—was described before by Belhadjer for a small group of very severely ill children hospitalised in intensive care units [14]. Due to having a broader group of MIS-C patients involved in the study, we could analyse the dynamics of LVEF changes among children who did not require intensive care. We observed that the real problem of left ventricular dysfunction in the acute phase of the disease might have been underestimated. This observation is in line with a previous report by Matsubara, who described the presence of cardiac strain abnormalities in children having normal LVEF [34]. In summary, acute heart failure observed in the course of MIS-C is transient, but the measurement of ventricular function in children with MIS-C should not rely on a single LVEF measure.

It has been already established that the picture of MIS-C depends on age, and older children have a higher risk for severe presentation [37,46,49]. In the context of cardiac involvement, Campanello showed that coronary artery involvement is more common in children aged less than 6 years while left ventricular dysfunction is more common in children aged at least 6 years of age [55]. Abrams also found that the risk for decreased cardiac function increased with each age group (0–5 vs. 5–12 vs. 12–18 years old) [49]. Our study is in line with these findings.

An increased hospitalisation incidence for patients with MIS-C followed the waves of COVID-19 in Poland [36,56]. The need for paediatric cardiologist evaluation exceeds the access in many Polish cities. Therefore, we looked for potential predictors of any echocardiographic findings and precise features, such as decreased LVEF, to facilitate decision making regarding which patient should be prioritised for paediatric cardiologist evaluation.

Previous studies revealed that increased concentrations of NT-proBNP correlated with worse left ventricular function and a higher risk of cardiogenic shock [49,57]. Raynor’s study showed that increased NT-proBNP correlated with the severity of left ventricular dysfunction and was a better marker than BNP [58]. On the other hand, Fridman found that higher troponin I, not NT-proBNP, was associated with decreased LVEF [59]. Campanello reported that fibrinogen concentration was increased in children with reduced LVEF [55]. Our results support the usefulness of all these laboratory cardiac markers, i.e., troponin, BNP and NT-proBNP, as laboratory features more common in children with contractility abnormalities. However, in looking for markers of any echo abnormalities, only troponin was identified to significantly more often be elevated in children with any echo changes as compared with children with a normal echo picture. Neither BNP nor NT-proBNP differed between these groups. We did not find significant differences in fibrinogen concentration between children with and without impaired ventricular function. Another finding of our study was that abnormalities in the laboratory tests results, such as in complete blood count results, inflammatory marker panels and sodium concentration, were significantly more common. Most importantly, these significant findings were already present at the time of patient admission. The broad availability of these simple tests makes them a very promising tool for the screening of children who are at risk of acute heart failure in the course of MIS-C, but further studies are definitely needed to evaluate their robustness as predictors.

While cases of takotsubo syndrome (TTS) in adults with COVID-19 have been described [60,61], data on this subject in the paediatric population are very scarce. Most of the documented cases of TTS in the paediatric population have been related to emotional experiences, psychiatric disorders or the use of psychoactive substances [28,62]. Severe disease can also be a stress factor for the body, resulting in strong activation of the adrenergic system. In the largest analysis of TTS in children so far, which included 153 cases aged up to 20 years, 22.3% of TTS cases occurred in patients with sepsis [28]. The immunological disorders observed in the course of MIS-C can be described as a cytokine storm [24,26,63,64]. MIS-C has a rapid course, often leading to the exhaustion of the body’s compensatory mechanisms. Such a strong inflammatory reaction can be seen as a sufficient trigger for the occurrence of TTS. Belhadjer mentioned an echocardiographic image resembling TTS in a patient with acute heart failure in the course of MIS-C [14]; we have also described a similar case before [29]. In this analysis, we extracted the cases with a description of apex rounding or other segmental contractility abnormality. Overall, 31 children had echo pictures that might have resembled TTS during hospitalisation. Interestingly, there were more such cases identified in the follow-up than in the initial echo, suggesting that such presentation is relatively late. Further studies are necessary to establish the connection between MIS-C and TTS.

## 5. Limitations

The relatively low number of available electrocardiography descriptions and inconsistencies in their descriptions did not allow us to include ECG analyses in this report. Diastolic dysfunction and right ventricular dysfunction echo measures were not included for the same reason. However, diastolic function assessment in children might be unreliable due to volume status, valve competency impact and poor interobserver agreement.

The MIS-C case report form did not include information on the temporality of the signs and symptoms presented by the patients; therefore, it was not clear whether these findings preceded, coincided or followed the development of echocardiological outcomes of interest. Therefore, we cannot rule out the possibility that these signs and symptoms may be markers of MIS-C, and not necessarily predictors of echo findings signalling potential cardiac dysfunction.

Our study also lacks records of long-term outcomes, and so we were not able to correlate our findings with specific clinical prognoses. However, this study was intended as exploratory research conducted to discover potential clinical markers, features or risk factors associated with MIS-C, to be later correlated with specific outcomes of interest. The strength of the study lies in the size of the cohort, as well as the depth of clinical variables, patient signs and symptoms investigated, especially in relation to echocardiological features for cardiac involvement.

## 6. Conclusions

Almost all children in our population had laboratory or echocardiographic features of cardiovascular involvement in the course of MIS-C. The incidence of left ventricular dysfunction might be underestimated if the assessment relies only on a single measurement of LVEF. The majority of children with acute heart failure improved significantly within a few days. Coronary artery abnormality was a relatively rare finding. We found simple clinical features were present more commonly in children with an impaired contractility as well as other cardiac findings that might facilitate decision making to assess the need for specialist paediatric cardiologist consultation. Due to the exploratory nature of this study, these findings should be confirmed in further association studies.

## Figures and Tables

**Figure 1 biomedicines-11-01251-f001:**
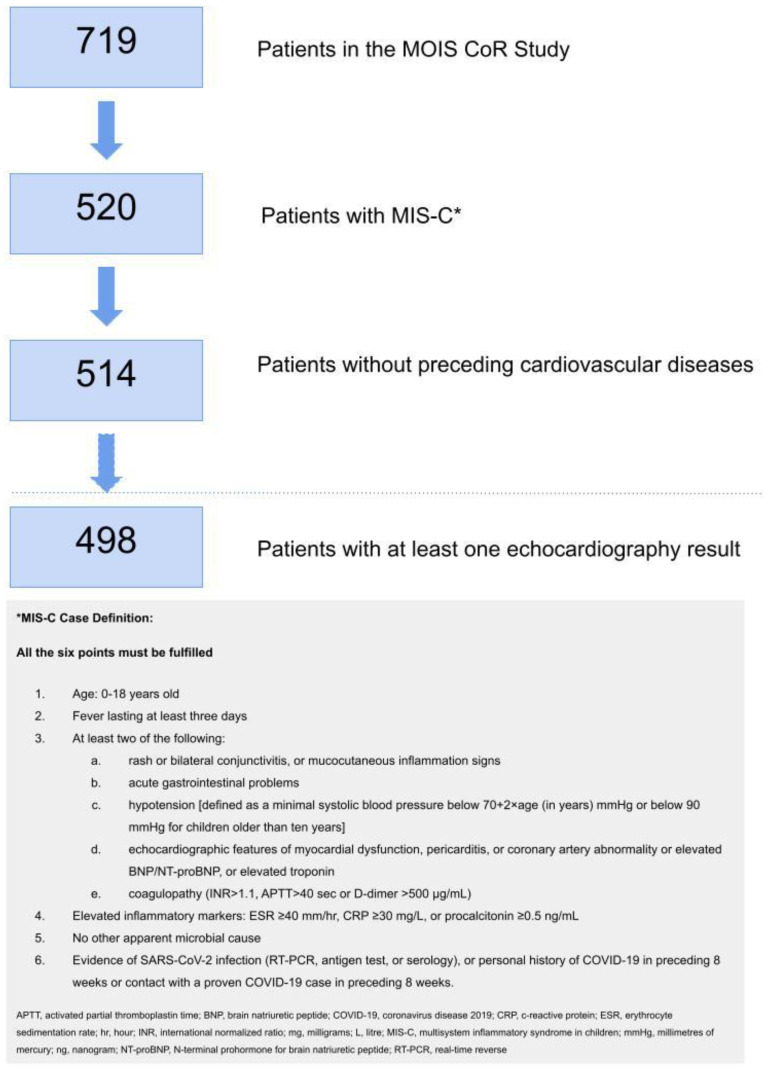
The flow chart for the study recruitment process and the MIS-C definition applied for the study. Study recruitment process.

**Figure 2 biomedicines-11-01251-f002:**
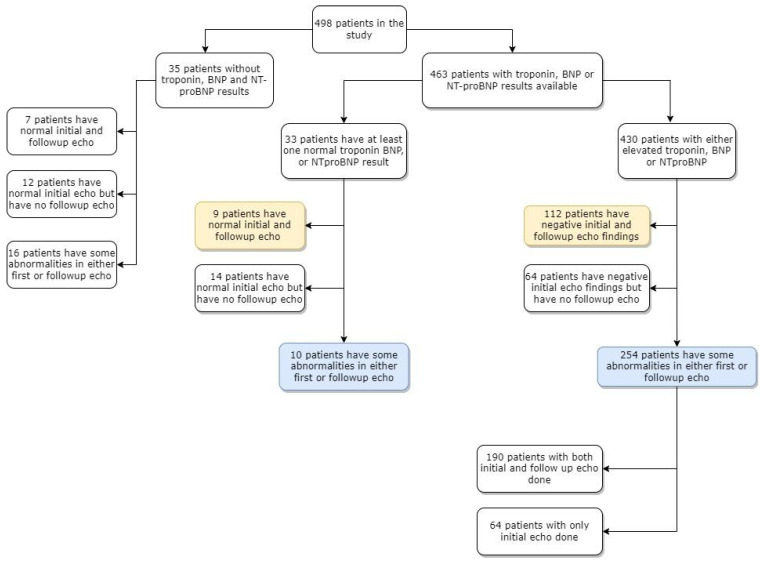
Clustering the study population based on availability of cardiac tests and results. Yellow colour indicates patients with normal echo descriptions and blue colour indicates patients with abnormalities on echo descriptions. Abbreviations: BNP, brain natriuretic peptide; echo, echocardiography; NT-proBNP N-terminal prohormone for brain natriuretic peptide.

**Figure 3 biomedicines-11-01251-f003:**
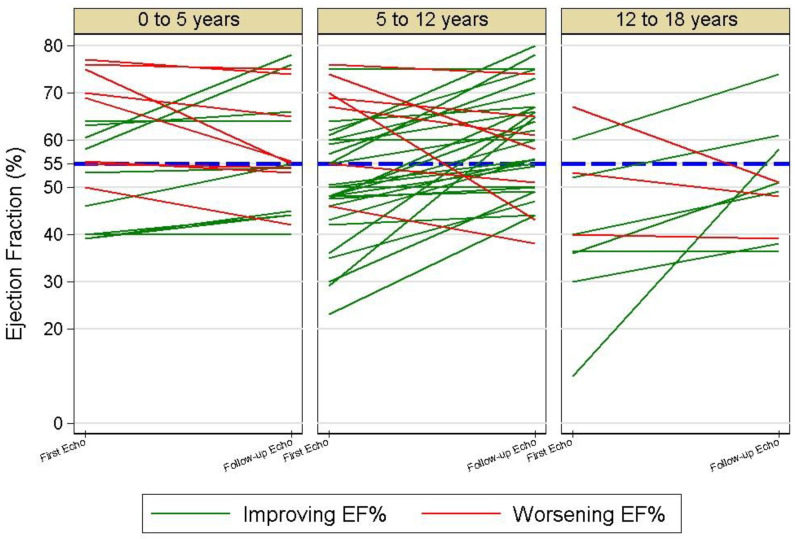
Left ventricular ejection fraction changes during the hospitalisation with the multisystem inflammatory syndrome in children (MIS-C) in age groups.

**Table 1 biomedicines-11-01251-t001:** The characteristics of the study group and multisystem inflammatory syndrome course by age group.

		Overall Cohort	<5 Years	5–11 Years	12–18 Years	*p*-Value
Counts (%) or Median (25th–75th percentile)		498 (100%)	137 (28%)	251 (50%)	110 (22%)	
**Demographical features**						
Male sex		314 (64%)	76 (56%)	159 (64%)	79 (73%)	**0.02**
Age		8.3 (4.7–11.6)	3.2 (1.9–4.2)	8.60 (6.7–10.5)	13.8 (13.0–15.2)	-
Race	White	496 (99.6%)	135 (99%)	251 (100%)	110 (100%)	0.12
	Asian	2 (0.4%)	2 (1%)	0 (0%)	0 (0%)	
BMI	kg/m^2^	16.6 (14.9–19.6)	15.2 (14.4–16.5)	16.1 (14.7–19.0)	20.6 (18.3–23.1)	**0.00**
BMI centiles		51.9 (21.1–80.7)	40.9 (17.4–77.0)	50.7 (20.6–82.6)	65.9 (30.8–83.9)	**0.01**
Obesity		33 (7%)	7 (6%)	18 (8.%)	8 (8%)	0.19
**Comorbidities ***						
None		447 (93%)	126 (95%)	223 (93%)	98 (91%)	0.47
**Signs and symptoms**						
Days of fever		7 (6–9)	7 (6–8)	7 (6–9)	7 (6–9)	0.12
Gastrointestinal symptoms		436 (89%)	110 (83%)	224 (90%)	102 (94%)	**0.03**
Mucocutaneous involvement		475 (96%)	130 (96%)	243 (97%)	102 (94%)	0.42
Upper respiratory symptoms		173 (37%)	48 (39%)	80 (33%)	45 (42%)	0.23
Lower respiratory symptoms		217 (46%)	46 (36%)	105 (44%)	66 (63%)	**0.00**
Neurological symptoms		383 (91%)	111 (93%)	188 (89%)	86 (92%)	0.41
Osteoarticular and muscle involvement		172 (37%)	32 (27%)	94 (39%)	46 (44%)	**0.02**
Systemic oedema		2 (2%)	0 (0%)	1 (2%)	1 (4%)	0.70
**Vital signs at peak of disease**						
AVPU other than A		61 (14%)	18 (15%)	28 (12%)	15 (15%)	0.71
Heartrate (beats/min)	Max	135 (120–150)	145 (130–160)	130 (120–141)	128 (110–140)	**0.00**
	Min	69 (57–81)	80 (68–96)	68 (56–80)	62 (50–70)	**0.00**
Prolonged CRT (>2 s)		53 (14%)	12 (12%)	25 (14%)	16 (20%)	0.30
Systolic blood pressure (mmHg)		85 (76–93)	85 (75–93)	84 (78–92)	84 (75–95)	0.59
Hypotension		206 (49%)	27 (25%)	118 (56%)	61 (62%)	**0.00**
Max. respiratory rate (breaths/min)		25 (20–30)	28 (22–37)	25 (20–30)	25 (20–30)	0.08
Min. SatO2 (%)		96 (93–97)	96 (94–98)	96 (93–98)	95 (92–97)	**0.02**
**Laboratory test results at the peak of disease**						
WBC max (10^3^/μL)		15.4 (11.1–20.6)	16.8 (12.3–21.8)	14.9 (10.9–20.3)	14.7 (11.5–20.0)	0.09
Neutrophils max (10^3^/μL)		10.1 (7.3–14.9)	9.6 (6.4–14.6)	9.9 (7.3–14.5)	12.3 (8.8–16.2)	**0.05**
Lymphocytes min (10^3^/μL)		1.0 (0.6–1.7)	1.7 (1.0–3.0)	0.9 (0.6–1.4)	0.6 (0.5–1.0)	**0.00**
Hb min (g/dL)		10.4 (9.5–11.2)	9.7 (9.0–10.4)	10.5 (9.7–11.2)	10.9 (10.1–11.9)	**0.00**
PLT min (10^3^/μL)		163 (109–228)	174 (111–265)	164 (110–224)	148 (107–203)	0.09
CRP max (mg/L)		169 (98–242)	139 (88–208)	169 (100–242)	209 (121–289)	**0.00**
ESR max (mm/h)		57.0 (35.0–77.0)	60.0 (36.0–72.0)	58.0 (35.0–81.0)	50.0 (33.0–77.0)	0.73
Fibrinogen max (g/L)		5.7 (4.6–7)	5.5 (4.5–6.7)	5.6 (4.5–6.9)	6.3 (5.3–7.3)	**0.03**
Procalcitonin max (ng/mL)		4.1 (1.4–12.8)	4.9 (2.2–10.4)	3.7 (1.3–12.3)	3.4 (1.1–13.4)	0.61
Ferritin max (ug/L)		387 (216–645)	270 (156–421)	413 (249–644)	511 (238–929)	**0.00**
Albumins min (g/dL)		2.9 (2.6–3.4)	2.9 (2.6–3.4)	2.9 (2.5–3.4)	3.0 (2.6–3.5)	0.29
Sodium min (mmol/L)		133 (130–135)	134 (132–136)	133 (130–135)	133 (129–135)	**0.03**
D-dimer max (mg/L)		3.7 (2.0–5.6)	3.8 (1.7–5.3)	3.7 (2.1–6.1)	3.5 (2.2–5.4)	0.51
eGFR min (mL/min/1.73 m^2^)		105 (85–126)	112 (90–137)	107 (88–125)	88 (65–118)	**0.00**
BNP max (pg/mL)		1070 (264–5245)	329 (87–1458)	1577 (342–5580)	2551 (448–11,480)	**0.00**
NT-proBNP max (pg/mL)		4744 (1462–11,479)	4737 (1166–15,005)	4136 (1592–9679)	6340 (2718–16,000)	0.14
Troponin elevated		97 (50%)	5 (16%)	53 (51%)	39 (64%)	**0.00**
**Echocardiography features**						
Any CAA		36 (11%)	17 (18%)	12 (7%)	7 (8%)	**0.04**
Any contractility		155 (41%)	27 (28%)	75 (40%)	53 (59%)	**0.00**
Any pericardial effusion		47 (13%)	13 (13%)	23 (14%)	11 (13%)	1.00
Any valvular insufficiency		190 (48%)	45 (43%)	104 (53%)	41 (45%)	0.15
Decreased LVEF (%)		132 (36%)	22 (23%)	65 (39%)	45 (51%)	**0.00**
LVEF in initial echo (%)		55.0 (46.0–64.0)	55.0 (46.0–64.0)	54.0 (46.0–63.2)	55.0 (46.0–65.0)	0.38
LVEF in follow-up echo (%)		55.8 (49.0–66.0)	55.0 (45.0–66.0)	58.0 (49.0–67.0)	51.0 (48.0–58.0)	0.12
**Management**						
PICU treatment		32 (7%)	3 (2%)	17 (7%)	12 (11%)	**0.02**
IVIG administered		447 (91%)	128 (94%)	223 (90%)	96 (91%)	0.31
GCS administered		344 (72%)	86 (64%)	170 (72%)	88 (81%)	**0.01**
ASA administered		438 (100%)	125 (100%)	224 (100%)	89 (100%)	**-**
Heparin administered		128 (38%)	28 (30%)	61 (37%)	39 (47%)	0.07

Abbreviations: AVPU, alert, verbal, pain, unresponsive; ASA, acetylsalicylic acid; BMI, body mass index; BNP, brain natriuretic peptide; CRP, C-reactive protein; CRT, capillary refill time; eGFR, estimated glomerular filtration rate; ESR, erythrocyte sedimentation rate; GCSs, glucocorticosteroids; Hb, haemoglobin; Hct, haematocrit; Il-6, interleukin 6; IQR, interquartile range; IVIGs, intravenous immunoglobulins; LDH, lactate dehydrogenase; LVEF, left ventricular ejection fraction; min, minutes; n, number; NT-proBNP, N-terminal prohormone for brain natriuretic peptide; PICU, paediatric intensive care unit; PLT, platelet count; s, seconds; SatO2, oxygen saturation; WBC, white blood count. * other than cardiovascular. Gastrointestinal symptoms encompassed nausea, vomiting, diarrhoea or abdominal pain; mucocutaneous involvement encompassed rash, erythema at BCG site, conjunctivitis, hands and feet erythema or oedema, digital peeling, inflammation of the oral cavity or cervical lymphadenopathy; upper respiratory symptoms encompassed coryza or sore throat; lower respiratory symptoms encompassed cough, breathing effort, chest pain and swallowing difficulty; neurological involvement encompassed meningeal signs, lethargy, seizures, headache, muscle hypotension, peripheral nerve paralysis, paresis, loss of smell or taste, photophobia, agitation or skin hyperesthesia; osteoarticular and muscle involvement encompassed arthritis, arthralgia or muscle pain. Max stands for the maximal value of the result obtained. Min stands for the minimal value of the result obtained. *p* values less than 0.05 are bolded.

**Table 2 biomedicines-11-01251-t002:** Changes between the baseline and follow-up echocardiography findings in patients with multisystem inflammatory syndrome in children (MIS-C) and two available echo descriptions.

Patients with Both Initial and Follow-up Echo (n = 337)				
Feature Counts (%) or Median (25th–75th Percentile)		Initial Echo	Follow-up Echo	*p*-Value
**Coronary artery abnormality ***				
None		315 (94.3%)	315 (94.3%)	1.00
Dilation		5 (1.5%)	6 (1.8%)	
Aneurysms		14 (4.2%)	13 (3.9%)	
**Contractility**				
Contractility described as abnormal		97 (29.0%)	63 (18.8%)	**0.00**
Heart apex rounding or segmental contraction disfunction		11 (3.3%)	20 (5.9%)	0.08
LVEF (%)	First	55.5 (47–64)	56.5 (49.5–68.5)	**0.02**
	Follow-up			
Decreased LVEF ^#^		87 (25.8%)	47 (13.9%)	**0.00**
**Valves**				
Any valve insufficiency described		102 (30.4%)	82 (24.4%)	**0.04**
**Pericardial effusion**				
Pericardial effusion		26 (7.7%)	15 (4.5%)	0.09

Abbreviations: CAA, coronary artery abnormality; Echo, echocardiography; EF, ejection fraction; IAoV, aortic valve insufficiency; IMV, mitral valve insufficiency; IPV, pulmonary valve insufficiency; ITV, tricuspid valve insufficiency. LVEF, left ventricular ejection fraction. # LVEF was evaluated using Simpson’s biplane method. * The worst instance was counted. *p* values less than 0.05 are bolded.

**Table 3 biomedicines-11-01251-t003:** Comparison between patients with and without echocardiographic abnormalities in the course of the multisystem inflammatory syndrome in children (MIS-C).

		Whole Cohort	Patients with No Abnormalities on First and Follow-Up Echo	Patients with Abnormalities on First or Follow-Up Echo	*p*-Value
Feature Counts (%) or Median (25th–75th percentile)		n = 498	n = 121	n = 264	
**Demographical features**					
Male sex	Male	314 (64%)	69 (58%)	175 (67%)	0.09
Age		8.3 (4.7–11.6)	8.3 (4.4–11.1)	8.7 (5.0–12.1)	0.16
BMI (kg/m^2^)		16.6 (14.9–20.0)	16.6 (14.8–19.4)	16.83813 (1503.8%)	0.10
Comorbidities		34 (7%)	7 (6%)	0.00 (0.00–0.00)	0.53
**Signs and symptoms**					
Days of fever		7 (6–9)	7 (6–9)	7 (6–8)	0.43
Gastrointestinal symptoms		436 (89%)	108 (91%)	25 (10%)	1.00
Mucocutaneous involvement		475 (96%)	117 (98%)	233 (90%)	0.51
Upper respiratory symptoms		173 (37%)	40 (36%)	9 (3%)	0.91
Lower respiratory symptoms		217 (46%)	45 (42%)	253 (97%)	0.09
Neurological symptoms		383 (91%)	91 (83%)	166 (65%)	**0.00**
Osteoarticular and muscle involvement		172 (37%)	44 (38%)	91 (35%)	0.91
Systemic oedema		2 (2%)	0 (0%)	124 (48%)	1.00
Breathing effort		101 (21%)	17 (15%)	70 (27.0%)	**0.01**
**Vital signs at admission**					
AVPU other than A		23 (5%)	2 (2%)	16 (7%)	0.07
Heartrate (beats/min)		120 (105–138)	120 (100–140)	124 (110–140)	0.14
Prolonged CRT (>2 s)		50 (13%)	15 (16%)	27 (13%)	0.59
Systolic blood pressure (mmHg)		99 (89–109)	101 (93–110)	98 (87–108)	**0.03**
Hypotension		64 (17%)	9 (9%)	46 (22.1%)	**0.01**
Respiratory rate (breaths/min)		20 (18–26)	20 (18–26)	22 (18–28)	**0.04**
SatO2 (%)		98 (96–99)	98 (96–99)	98 (97–99)	0.61
**Laboratory test results at admission**					
WBC (10^3^/μL)		9.6 (6.7–13.0)	9.7 (6.5–12.7)	9.71 (6.65–13.66)	0.87
Neutrophils (10^3^/μL)		7.6 (4.9–10.4)	7.8 (5.0–10.0)	7.9 (4.8–11.0)	0.90
Lymphocytes (10^3^/μL)		1.0 (0.7–1.8)	1.2 (0.7–1.8)	0.9 (0.6–1.5)	0.15
Hb (g/dL)		11.7 (10.7–12.7)	11.7 (10.6–12.6)	11.7 (10.7–12.7)	0.91
PLT (10^3^/μL)		179 (130–248)	190 (139–250)	175 (122–238)	0.21
CRP (mg/L)		144 (84–206)	141 (84–188)	150 (87–223)	0.12
Procalcitonin (ng/mL)		2.8 (1.1–7.6)	1.8 (0.8–5.1)	3.5 (1.3–12.8)	**0.00**
Ferritin (ug/L)		331 (198–567)	316 (177–532)	401 (212–708)	0.07
Albumins (g/dL)		3.4 (3.0–3.8)	3.4 (2.8–3.7)	3.4 (2.9–3.7)	0.88
Sodium (mmol/L)		134 (132–136)	135 (132–137)	133 (131–136)	**0.00**
D-dimer (mg/L)		2.6 (1.5–4.5)	2.4 (1.4–4.4)	2.9 (1.7–4.8)	0.08
eGFR (mL/min/1.73 m^2^)		109 (86–132)	113 (95–135)	104 (82–132)	0.23
BNP (pg/mL)		287 (114–1006)	449 (155–865)	302 (94–1657)	0.36
NT-proBNP (pg/mL)		2176 (486–7426)	2010 (651–5372)	3399 (641–10,317)	0.08
Troponin elevated		63 (26%)	12 (16%)	46 (35%)	**0.00**
**Management**					
PICU treatment		32 (7%)	6 (5%)	25 (10%)	0.16
IVIG administered		447 (91%)	111 (93%)	253 (97%)	0.12
GCS administered		344 (72%)	82 (72%)	202 (78%)	0.19

Abbreviations: AVPU, alert, verbal, pain, unresponsive; BMI, body mass index; BNP, brain natriuretic peptide; CRP, C-reactive protein; CRT, capillary refill time; eGFR, estimated glomerular filtration rate; GCSs, glucocorticosteroids; Hb, haemoglobin; IVIGs, intravenous immunoglobulins; NT-proBNP, N-terminal prohormone for brain natriuretic peptide; PICU, paediatric intensive care unit; PLT, platelet count; s, seconds; SatO2, oxygen saturation; WBC, white blood count. Other than cardiovascular. Gastrointestinal symptoms encompassed nausea, vomiting, diarrhoea or abdominal pain; mucocutaneous involvement encompassed rash, erythema at BCG site, conjunctivitis, hands and feet erythema or oedema, digital peeling, inflammation of the oral cavity or cervical lymphadenopathy; upper respiratory symptoms encompassed coryza or sore throat; lower respiratory symptoms encompassed cough, breathing effort, chest pain and swallowing difficulty; neurological involvement encompassed meningeal signs, lethargy, seizures, headache, muscle hypotension, peripheral nerve paralysis, paresis, loss of smell or taste, photophobia, agitation or skin hyperesthesia; osteoarticular and muscle involvement encompassed arthritis, arthralgia or muscle pain. Max stands for the maximal value of the result obtained. Min stands for the minimal value of the result obtained. *p* values less than 0.05 are bolded.

**Table 4 biomedicines-11-01251-t004:** Comparison between children with and without coronary artery abnormalities (CAAs), contractility abnormalities and pericardial effusion in the course of the multisystem inflammatory syndrome in children.

		Patients with CAAs at Any Time	Patients with No CAAs at Any Time	*p*	Patients with Contractility Abnormalities at Any Time	Patients with No Contractility Abnormalities at Any Time	*p*	Patients with Pericardial Effusion at Any Time	Patients with no Pericardial Effusion at Any Time	*p*
Feature Counts (%) or Median (25th–75th percentile)		n = 36	n = 307		n = 155	n = 219		n = 47	n = 305	
**Demographical features**										
Male sex		25 (69%)	186 (61%)	0.37	105 (69%)	131 (60%)	0.08	35 (76%)	183 (60%)	**0.05**
Age (years)		5 (3–12)	9 (5–12)	**0.03**	10 (6–13)	7 (4–11)	**0.00**	8.2 (4.6–11.5)	8.3 (4.9–11.9)	0.86
BMI (kg/m^2^)		16.5 (15.4–18.1)	16.6 (14.9–20.0)	0.93	17.7 (15.6–20.9)	16.0 (14.7–18.8)	**0.00**	17.3 (15.3–20.5)	16.4 (14.9–19.6)	0.11
**Comorbidities**										
None		34 (97%)	268 (92%)	0.34	138 (91%)	193 (93%)	0.69	45 (98%)	266 (91%)	0.23
**Signs and symptoms**										
Days of fever		7 (7–9)	7 (6–9)	0.22	7 (6–8)	7 (7–9)	**0.03**	7 (6–9)	7 (6–9)	0.97
Gastrointestinal symptoms		29 (83%)	276 (91%)	0.12	139 (92%)	194 (90%)	0.58	42 (89%)	270 (90%)	0.80
Mucocutaneous involvement		33 (94%)	294 (97%)	0.36	148 (96%)	209 (97%)	0.78	45 (96%)	290 (96%)	0.69
Upper respiratory symptoms		13 (38%)	111 (38%)	1.00	49 (32%)	78 (38%)	0.27	18 (40%)	105 (36%)	0.62
Lower respiratory symptoms		17 (50%)	144 (50%)	1.00	82 (54%)	90 (45%)	0.09	28 (61%)	136 (48%)	0.11
Neurological symptoms		24 (86%)	249 (90%)	0.51	124 (95%)	169 (87%)	**0.01**	38 (97%)	242 (89%)	0.15
Osteoarticular and muscle involvement		12 (36%)	118 (41%)	0.71	51 (34%)	86 (42%)	0.19	17 (38%)	117 (40%)	0.87
Systemic oedema		0 (0%)	2 (3%)	1.00	2 (6%)	0 (0%)	0.21	0 (0%)	2 (3%)	1.00
Particular symptoms										
Arthritis		5 (14%)	11 (4%)	**0.02**	5 (3%)	12 (6%)	0.32	3 (7%)	13 (4%)	0.46
Conjunctivitis		22 (61%)	242 (81%)	**0.02**	122 (80%)	168 (79%)	0.90	28 (61%)	243 (81%)	**0.00**
Eyelid swelling		1 (7%)	34 (36%)	**0.03**	32 (39%)	15 (26%)	0.15	9 (47%)	31 (31%)	0.19
Rhinitis		8 (23%)	32 (11%)	0.05	13 (8%)	24 (12%)	0.38	6 (13%)	32 (11%)	0.63
Oral inflammation		20 (57%)	210 (71%)	0.12	104 (69%)	144 (69%)	1.00	26 (55%)	207 (71%)	**0.04**
Breathing effort		9 (25%)	70 (24%)	0.84	46 (30%)	40 (19.%)	**0.02**	16 (35%)	66 (22%)	0.09
Chest pain		3 (9%)	49 (17%)	0.32	33 (22%)	22 (11%)	**0.01**	7 (16%)	45 (16%)	1.00
Abdominal pain		22 (67%)	247 (83%)	**0.03**	129 (85%)	167 (80%)	0.17	35 (76%)	240 (82%)	0.42
Nausea		14 (41%)	188 (63%)	**0.02**	91 (62%)	130 (61%)	1.00	28 (61%)	180 (61%)	1.00
Diarrhoea		23 (64%)	181 (60%)	0.72	87 (57%)	127 (59%)	0.75	20 (43%)	184 (61%)	**0.02**
**Vital signs at admission**										
Systolic blood pressure (mmHg)		102 (87–119)	100 (89–109)	0.32	92 (84–104)	101 (92–110)	**0.00**	98 (89–103)	100 (89–110)	0.62
Hypotension		3 (11%)	42 (17%)	0.59	40 (32%)	15 (9%)	**0.00**	5 (13%)	41 (17%)	0.65
Respiratory rate (breaths/min)		24 (20–30)	20 (18–26)	0.34	22 (20–30)	20 (18–25)	**0.00**	22 (19–28)	20 (18–26)	0.13
SatO2 (%)		98 (97–99)	98 (96–99)	0.59	98 (96–99)	98 (97–99)	**0.04**	97 (96–98)	98 (96–99)	**0.01**
**Laboratory test results at admission**										
Lymphocytes (10^3^/μL)		1.3 (0.7–2.7)	1.0 (0.7–1.6)	**0.00**	0.8 (0.6–1.2)	1.2 (0.7–2.0)	**0.00**	1.0 (0.7–1.7)	1.0 (0.7–1.7)	0.69
PLT (10^3^/μL)		191 (151–325)	172 (122–244)	**0.01**	167 (122–216)	186 (127–273)	**0.00**	158 (114–239)	180 (127–248)	0.89
CRP (mg/L)		121 (55–173)	145 (91–205)	0.07	171 (102–240)	137 (84–188)	**0.00**	164 (86–232)	143 (87–196)	0.23
Procalcitonin (ng/mL)		3.2 (0.9–6.8)	2.8 (1.1–8.6)	0.78	4.6 (1.7–16.0)	2.0 (0.8–6.2)	**0.00**	3.8 (1.3–10.6)	2.5 (1.0–8.3)	0.23
Ferritin (ug/L)		354 (150–771)	341 (197–567)	0.87	435.70 (235.10–760.20)	318 (181–532)	**0.015**	368 (237–581)	336 (186–576)	0.67
Sodium (mmol/L)		136 (132–138)	134 (131–136)	0.10	133 (130–136)	134 (132–136)	**0.00**	135 (131–136)	134 (132–136)	0.67
eGFR (mL/min/1.73 m^2^)		100 (83–130)	112 (88–134)	0.30	99 (78–120)	116 (95–137)	**0.00**	98 (80.0–140)	110 (89–134)	0.96
BNP (pg/mL)		138 (39–695)	578 (143–1381)	**0.03**	1197 (328–2814)	211 (135–798)	**0.01**	818 (660–2814)	282 (135–1012)	0.22
NT-proBNP (pg/mL)		387 (220–4956)	2845 (651–7791)	**0.04**	5053 (1183–13,597)	1589 (424–5381)	**0.00**	3058 (368–15,995)	2128 (544–7400)	0.58
Troponin elevated		5 (31%)	43 (25%)	0.56	35 (49%)	18 (15%)	**0.00**	8 (27.6%)	42 (269%)	0.82
**Management**										
PICU treatment		3 (8%)	24 (8%)	1.00	20 (13%)	10 (5%)	**0.00**	7 (15%)	22 (7%)	0.08
IVIG administered		34 (97%)	288 (94%)	1.00	153 (99%)	202 (93%)	**0.01**	46 (98%)	287 (957%)	0.71
GCS administered		26 (74%)	219 (74%)	1.00	129 (85%)	143 (68%)	**0.00**	34 (76%)	216 (74%)	0.86

Abbreviations: AVPU, alert, verbal, pain, unresponsive; ASA, acetylsalicylic acid; BMI, body mass index; BNP, brain natriuretic peptide; CAA, coronary artery abnormality; CRP, C-reactive protein; CRT, capillary refill time; eGFR, estimated glomerular filtration rate; ESR, erythrocyte sedimentation rate; GCSs, glucocorticosteroids; Hb, haemoglobin; Hct, haematocrit; Il-6, interleukin 6; IQR, interquartile range; IVIG, intravenous immunoglobulins; LDH, lactate dehydrogenase; min, minutes; n, number; NT-proBNP, N-terminal prohormone for brain natriuretic peptide; PICU, paediatric intensive care unit; PLT, platelet count; s, seconds; SatO2, oxygen saturation; WBC, white blood count. The white and grey colours were used to divide the independent subgroups presented. *p* values less than 0.05 are bolded.

## Data Availability

The data that support the findings of this study are available from the corresponding author, K.M.L., upon reasonable request.

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
