# Peer review of "Cardiac Involvement in Patients with Multisystem Inflammatory Syndrome in Children (MIS-C) in Poland"

_biomedicines, 2023, doi:10.3390/biomedicines11051251_

Round 1

Reviewer 1 Report

Dear Editor and Authors,

Thank you for asking me to review this work by Dr. Ludwikowska and colleagues from the Wroclaw Medical University in Poland titled: "Cardiac involvement in patients with multisystem inflammatory syndrome in children (MIS-C) in Poland".

Overall this is a very interesting study, looking at cardiac sequelae and dysfunction (transient or not) in children who have developed MIS-C post Covid-19 infection.

The study is quite sound methodologically and despite its retrospective nature has quite clear parameters and exclusion/inclusion criteria. It is multi-institutional covering the whole of Poland and quite thorough. The analysis is quite comprehensive and complete albeit sometimes it feels quite cluttered and difficult to discern at certain places (especially the tables).

It is well written and presented and quite clear for the reader to understand with only minor language editing required. 

My only comment is (as previoudly mentioned) that at a certain point it feels that too much information is given!! i.e. blood results and vital signs ect. I am uncertain how much these are needed to demonstrate the impact of MIS-C on cardiac functionality. Nevertheless, I will leave it to the authors to correct it if they agree with me!!

Thank you for giving me the oportiunity to review this work, wishing all well.

Author Response

Dear Reviewer,

We appreciate your opinion on our manuscript and we are very happy that you found it valuable. As you have suggested, we have revised the methods and results and reorganized the results sections, mostly the tables, and moved some data into the supplements to underline the most important results and to improve the clarity of the manuscript. 

We hope you find the updated version of our manuscript even more valuable.

Reviewer 2 Report

I reviewed with interest the manuscript of Ludwikowska et al. "Cardiac involvement in patients with multisystem inflammatory syndrome in children (MIS-C) in Poland". The authors in their article analyzed the data of the national register of patients with multisystem inflammatory syndrome in children, which made it possible to include a large number of patients. Therefore, the analysis of the incidence of Cardiac involvement in these patients is of interest and contains some new scientific data.

However, when reviewing, I had questions and comments to which I would like to receive answers from the authors.

1. The list of references is formatted incorrectly, the links are presented as if they were on the INTERNET sites. In fact, publications in journals are still mostly cited, so the references should be formatted appropriately. For example - one of the recent publications (Orlando L et al).

2. The Introduction section usually ends with the formulated goal of the study, in this manuscript there is one more paragraph after the goal, it should be placed before the goal of the study. If the authors deem it necessary, the phrase about the study of signs of takotsubo syndrome can be added to the purpose of the study.

3. Many questions on the Statistical methods section.

- no information - whether the quantitative data were checked for normal distribution (and if it was carried out, then by what method)

- when describing quantitative data, the authors used the median and interquartile range (which is usually used for non-parametric distribution), and to compare these data, Student's t-test and ANOVA test (which are used for normal distribution). It is not clear why the authors use these mutually exclusive techniques. That is, with a non-parametric distribution, the Mann-Whitney test and the Kruskal-Wallis test should be used.

- it is not clear what the p-values in Table 1 refer to. If these are the results of a comparison of three groups, then why are inter-group comparisons between each two groups not given?

- the authors identify many subgroups and compare them in many ways. How did the authors take into account the effect of multiple comparisons in doing so, did they adjust p-values for these cases?

4. In my opinion, it would be more appropriate to use multiple logistic regression methods to characterize the factors associated with certain cardiac manifestations of multisystem inflammatory syndrome in children

5. Section Discussion should begin with the main results obtained by the authors, and then compare them with literature data. Apparently, the third paragraph of this section should have been the initial one.

6. In the Summary there is no decoding of the abbreviation CAA.

References:

1.       Orlando, L.; Bagnato, G.; Ioppolo, C.; Franzè, M.S.; Perticone, M.; Versace, A.G.; Sciacqua, A.; Russo, V.; Cicero, A.F.G.; De Gaetano, A.; et al. Natural Course of COVID-19 and Independent Predictors of Mortality. Biomedicines 2023, 11, 939. https://doi.org/10.3390/biomedicines11030939

Author Response

Dear Reviewer,

Thank you very much for your time and effort to revise our manuscript. We have applied the corrections. We believe that our manuscript was improved and has gained clarity thanks to your suggestions. 

  1. The list of references is formatted incorrectly, the links are presented as if they were on the INTERNET sites. In fact, publications in journals are still mostly cited, so the references should be formatted appropriately. For example - one of the recent publications (Orlando L et al).

Response: The citations style has been corrected as suggested 

  1. The Introduction section usually ends with the formulated goal of the study, in this manuscript there is one more paragraph after the goal, it should be placed before the goal of the study. If the authors deem it necessary, the phrase about the study of signs of takotsubo syndrome can be added to the purpose of the study.

Response: The introduction has been rearranged to follow the advised order. TTS signs investigation has been included in the study aims. 

  1. Many questions on the Statistical methods section.

- no information - whether the quantitative data were checked for normal distribution (and if it was carried out, then by what method)

Response: 

  • Thank you for all the advice on statistics. Methods and results have been improved significantly. The statistics used have been verified and completed according to the needs and type of the data. Our data has been checked for normal distribution using k-density and histograms. Most of the variables followed a normal distribution, while some variables showed skewness and kurtosis of expected levels. All the information on the statistical process is described in the corrected statistical methods section now. 

- when describing quantitative data, the authors used the median and interquartile range (which is usually used for non-parametric distribution), and to compare these data, Student's t-test and ANOVA test (which are used for normal distribution). It is not clear why the authors use these mutually exclusive techniques. That is, with a non-parametric distribution, the Mann-Whitney test and the Kruskal-Wallis test should be used.

Response: 

  • Although most of our variables followed a fairly normal distribution with a low degree of skewness and kurtosis, we considered representing all the data with the median and IQR values in our tables as medians are a better representative of the central tendency and will be less affected by any outliers found in some of our variables (examples: lymphocyte counts, NT-proBNP). Fisher’s exact test was used to assess independence between categorical variables. ANOVA was used for the continuous variables, as our data were fairly normal, and ANOVA is also known not to be sensitive to sensitive to moderate deviations from normality. Additionally, the statistical software used calculated Bartlett's Test along with the one-way ANOVA to check if variances were equal across groups or samples, and this assumption was not violated.

- it is not clear what the p-values in Table 1 refer to. If these are the results of a comparison of three groups, then why are inter-group comparisons between each two groups not given?

Response: 

  • In Table 1, a one-way ANOVA was used to compare our three major age groups (<5 years, 5-11 years, and 12-18 years) to see if the population were in any way statistically different from each other. We performed this analysis by these three groups as we suspect there may be varying physiologies and pathologies between pre-schoolers, school-going children, and adolescents, as well as differing presenting symptoms or complaints (eg: due to being verbal/ non-verbal). Most other studies do not have this advantage; hence we decided to present and explore our cohorts by these groupings. This is similar to what we had done in our previous paper (https://www.nature.com/articles/s41598-021-02669-2). The p-value supplied by the ANOVA does not tell us which specific groups are statistically significantly different from each other, only that at least two groups were. This was our intention, however, as we mean to do preliminary hypothesis testing for future studies. The focus of our paper is on echographic features and progression/prognoses of cardiac involvement in children with MIS-C. The presented results indicate the direction for further investigation of symptomatology and clinical features in-depth as suggested.

- the authors identify many subgroups and compare them in many ways. How did the authors take into account the effect of multiple comparisons in doing so, did they adjust p-values for these cases?

Response: 

  • All comparisons done within our study are between two groups or outcomes appearing at two-time points as described in the tables (eg: patients with versus without contractility abnormalities at any time; first echo results between patients with normal versus reduced LVEF patients, comparisons between initial and follow-up echography results), except for Table 1, in which we look at variables across our 3 age groups. As mentioned, Table 1 is a preliminary test meant for hypothesis generation in service to detailed consideration in a future paper.
  • To clarify, as each of the comparisons was performed independently, the statistical results of one table are not related to the results of another. Statistical methods were used depending on the overall nature of the data. As we performed preliminary investigations on our data, there is no risk of confounding variables impacting the results of our outcomes.
  1. In my opinion, it would be more appropriate to use multiple logistic regression methods to characterise the factors associated with certain cardiac manifestations of multisystem inflammatory syndrome in children

Response: Thank you for this suggestion. We find this study as a preliminary study aimed at precise characteristics of cardiac involvement in MIS-C. We will continue working to investigate the risk factors and predictors using logistic regression models. However, we think that it would be overwhelming with the number of results and aims in this manuscript. Such a concern has been already raised by the reviewers. 

  1. Section Discussion should begin with the main results obtained by the authors, and then compare them with literature data. Apparently, the third paragraph of this section should have been the initial one.

Response: Thank you for this suggestion. We have moved the third paragraph so the discussion on actual cardiac involvement starts with it now. We assume that not all the readers would be familiar with the MIS-C characteristics. Therefore, we believe that beginning the discussion with a short description of MIS-C (and at the same time a confirmation that our cohort was representative) is the best starting point for the further discussion.  

  1. In the Summary there is no decoding of the abbreviation CAA.

Response: Abbreviation expanded

Reviewer 3 Report

Ludwikowska et al. Reported on respectable cohort of MIS with cardiac involvement in pediatric population. The paper is significant as the cohort is sufficiently big to infer relevant clinical conclusions for these patients.

I have several comments that need to be properly addressed before further consideration of this manuscript:

1. Why didn’t the authors measured GLS, concerning its superiority in detecting systolic dysfunction, especially, when searching for subtle changes?

2. Statistical methods should not include number of patients in groups.

3. methods section is insufficently described, especially in regards to what exactly was assessed and in which way. In addition, it should be clearly defined what is considered a disorder. For instance, neurological disorders, which are reported in as much as 95% of patients.

4. Table 1. Please present data as n (%) and median (25th-75th percentile) as the table is hary readable now.

5. In consideration of low sensitivity/specificity of TTE in coronary artery assessment, I wonder if authors further confirmed dilations/aneurysms using CT angio

6. Table 2 and 3 should be one table (with comparisons). What do p values represent in the part of table 3 concerning valvular oathology, it seems like it is one p value for all data. That is inappropriate. Benign valvular regurgitations should be reported, but their benign nature (and lack of connection with MIS, especially TVI) should be addressed. 

7. In general tables are too big and very messy. The authors should address the main findings and transfer the rest in supplementary data; especially Table 5.

Author Response

Dear Reviewer, 

thank you for your review. 

  1. Why didn’t the authors measured GLS, concerning its superiority in detecting systolic dysfunction, especially, when searching for subtle changes?

Response: Thank you for this point. We agree with the Reviewer that echocardiographic global longitudinal strain (GLS) is superior when compared with conventional ejection fraction (EF) in detecting subtle changes in left ventricular (LV) function. However, the GLS reporting was not included in the study protocol. The echocardiographic examinations were provided from multiple sites in Poland, it was performed on different devices, and not all of them had access to GLS, therefore it would be hard to standardize such parameter, and potentially high number of missing data not allowed to include them. We used the most repetitive and universal features included in the reports. 

  1. Statistical methods should not include number of patients in groups.

Response: 

We corrected statistical methods in line with your advice as well as other reviewers’ suggestions

  1. methods section is insufficiently described, especially in regards to what exactly was assessed and in which way. In addition, it should be clearly defined what is considered a disorder. For instance, neurological disorders, which are reported in as much as 95% of patients.

Response:

We have improved significantly our methods section. We also provided information on signs and symptoms included in the grouped symptoms, ie neurological. This information is provided under the tables with such results now. 

  1. Table 1. Please present data as n (%) and median (25th-75th percentile) as the table is hary readable now.

Response: 

We have provided 25th-75th percentile in all the continuous data results presented 

  1. In consideration of low sensitivity/specificity of TTE in coronary artery assessment, I wonder if authors further confirmed dilations/aneurysms using CT angio

Response: CT angio was not used to confirm CAA for several reasons. Most importantly, the AHA recommendations for Kawasaki disease were applied - as cited below, CTA is not routinely advised in an acute phase of the disease, as the risk-benefit ratio of such a procedure for children is doubtful. Our data come only from the hospitalization period during the acute phase of the disease. Secondly, access to the CTA or CMRI imaging for children was limited in most pediatric centers where MIS-C patients were hospitalized.  

McCrindle BW, Rowley AH, Newburger JW, Burns JC, Bolger AF, Gewitz M, Baker AL, Jackson MA, Takahashi M, Shah PB, Kobayashi T, Wu M-H, Saji TT, Pahl E; on behalf of the American Heart Association Rheumatic Fever, Endocarditis, and Kawasaki Disease Committee of the Council on Cardiovascular Disease in the Young; Council on Cardiovascular and Stroke Nursing; Council on Cardiovascular Surgery and Anesthesia; and Council on Epidemiology and Prevention. Diagnosis, treatment, and long-term management of Kawasaki disease: a scientific statement for health professionals from the American Heart Association. Circulation. 2017;135:e927–e999. doi: 10.1161/CIR.0000000000000484.: Transesophageal echocardiography, invasive angiography, CMRI, and CTA can be of value in the assessment of selected patients but are not routinely indicated for diagnosis and management of the acute illness. Invasive angiography is rarely performed during the acute illness. Transesophageal echocardiography, CTA, and CMRI can be useful for the evaluation of older children and adolescents in whom visualization of the coronary arteries with transthoracic echocardiography is inadequate.153,154 Evaluation of potential aneurysmal involvement in other arterial beds can be assessed with CMRI, CTA, and, rarely, invasive angiography, but such assessment is best performed after recovery from the acute illness, and usually for patients with severe coronary artery involvement or symptoms or signs, such as the presence of a pulsatile axillary mass.”

  1. Table 2 and 3 should be one table (with comparisons). What do p values represent in the part of table 3 concerning valvular pathology, it seems like it is one p value for all data. That is inappropriate. Benign valvular regurgitations should be reported, but their benign nature (and lack of connection with MIS, especially TVI) should be addressed. 

Response: 

Table 2 has been shifted into the supplementary material as it is less valuable and only complements the whole picture of cardiac involvement. Table 3 (now presented as Table 2) is the actual reliable table to present the comparison of the changes observed. The p-value is presented now in a more clear way - assigned to the single result it concerns. 

  1. In general tables are too big and very messy. The authors should address the main findings and transfer the rest in supplementary data; especially Table 5.

Response: The tables have been rearranged and limited to present the most important findings, and supplementary data have been provided to include all the data. 

Round 2

Reviewer 2 Report

The authors did a great job of correcting the text of the manuscript, responded to all my comments. I have no other comments.

Reviewer 3 Report

No further comments.